# Potential Clinical Applications of Pro-Resolving Lipids Mediators from Docosahexaenoic Acid

**DOI:** 10.3390/nu15153317

**Published:** 2023-07-26

**Authors:** María Paz Beyer, Luis A. Videla, Camila Farías, Rodrigo Valenzuela

**Affiliations:** 1Department of Nutrition, Faculty of Medicine, University of Chile, Santiago 8380000, Chile; m.pazbeyer@gmail.com (M.P.B.); camila.farias.castro@gmail.com (C.F.); 2Molecular and Clinical Pharmacology Program, Institute of Biomedical Sciences, Faculty of Medicine, University of Chile, Santiago 7810000, Chile; lvidela1944@gmail.com

**Keywords:** docosahexaenoic acid, specialized pro-resolving mediators, anti-inflammation, resolvins series D, protectins, maresins

## Abstract

Docosahexaenoic acid (C22:6*n*-3, DHA) is the precursor of specialized pro-resolving lipid mediators (SPMs), such as resolvin, protectin, and maresin families which have been considered therapeutic bioactive compounds for human health. Growing evidence indicates that DHA and SPMs are beneficial strategies in the amelioration, regulation, and duration of inflammatory processes through different biological actions. The present review discusses the reported therapeutic benefits of SPMs on various diseases and their potential clinical applications.

## 1. Introduction

Docosahexaenoic acid (C22:6*n*-3; DHA) is a long-chain polyunsaturated fatty acid (LCPUFA) of the *n*-3 series [1]. Human beings have a limited capacity for DHA synthesis from its essential precursor; therefore, it is necessary to provide a DHA supply through diet [2]. The main dietary sources of DHA include marine foods, such as fish (mackerel, salmon, herring, tuna, sardine, among others) and a variety of seafoods and microalgae [3]. The Food and Agriculture Organization (FAO-2010) advises a dosage of 250 mg/day of eicosapentaenoic acid (C20:5*n*-3; EPA) plus DHA for adult men and not pregnant or not lactating women [4], and the recommendation of the World Health Organization (WHO-2003) amounts to 200–500 mg/day of EPA + DHA for adults [5].

DHA is synthesized from dietary α-linolenic acid (C18.3*n*-3; ALA) by a complex enzymatic process of desaturation and elongation reactions predominantly occurring in the endoplasmic reticulum of the liver, excluding the last oxidation step taking place in the peroxisomes (Figure 1) [6]. The first reaction, catalyzed by delta-6 desaturase (Δ6-D) and producing stearidonic acid (C18:4*n*-3; SDA), is considered as the rate-limiting step in DHA formation [7]. SDA is transformed into eicosatetraenoic acid (C20:4*n*-3; ETA) by the action of elongase-5, followed by the desaturation of ETA into EPA by Δ5-D, and the successive elongations via elongases-5,2 that convert EPA into docosapentaenoic acid (C22:5*n*-3; DPA) and then to tetracosapentaenoic acid (C24:5*n*-3; TPA) [8], which is desaturated by Δ6-D forming tetracosahexaenoic acid (C24:6*n*-3; THA) (Figure 1). Finally, THA is subjected to peroxisomal oxidation leading to DHA production (Figure 1) [8]. Interestingly, the conversion of EPA to produce DHA is slower than that of ALA into EPA [9].

Current information supports the contention that DHA is an important bioactive compound for human health at the different periods of life. Related research has linked DHA with premature birth [10], cognitive function in early childhood [11,12], prevention of cardiovascular diseases [13,14], cognitive performance improvement in healthy young adults [15], and a significantly decreased risk of incident of age-related macular degeneration in women [16]. Among DHA’s actions, the diminution of inflammation and its resolution stands out, having a beneficial role in various acute and chronic diseases [17]. All these effects are performed by DHA itself or by its bioactive metabolites, namely the specialized pro-resolving mediators (SPMs) [18], including D-series resolvins, protectins and maresins, the cysteinyl-conjugated SPMs, and the recently identified SPMs from DPA [19]. This review addresses only the resolvins, protectins, and maresins, due to the definitive information on these mediators available at the present time.

The SPMs are formed through enzymatic or non-enzymatic pathways. The enzymatic routes involve (i) cyclooxygenases (COXs), (ii) lipoxygenases (LOXs), and (iii) cytochrome P450 mixed-function oxidase [20]. The non-enzymatic synthesis is mediated by free radical-related peroxidation processes [21], leading to the formation of different pro-resolving derivates of EPA including hydroxyeicosapentaenoic acids (HEPEs), such as 11-HEPE and 18-HEPE, among others [21,22].

SPM biosynthesis is altered in different diseases, suggesting that derangement of the endogenous pathways may be considered as an etiologic factor [23], since a diminished SPM synthesis can lead to uncontrolled inflammation with noxious results [24]. In this context, SPMs attenuate inflammatory processes in numerous diseases, including the post-stroke cognitive impairment [25], Alzheimer’s disease [26], sepsis [27], localized aggressive periodontitis [28], among others.

## 2. Methods

The search was carried out using the following key words: pro-resolving lipid mediators, maresin, resolvin, and protectin, by means of the PubMed, Science Direct, and Web of Science. Additional articles were identified in original publications from 2010 onwards, including in vitro, animal, and human trials analyzing the possible clinical applications of the DHA-derived mediators.

## 3. Synthesis and Metabolism of DHA-Derived Lipid Mediators

DHA is the precursor of several families of molecules that favor inflammation resolution, like D-series resolvins, protectins and maresins, each one of them exhibiting unique structures (Figure 2), receptors, and actions [29]. The enzymatic synthesis of SPMs involves complex processes that begin with the desesterification of membrane phospholipids, mainly by the action of phospholipase A_2_ [20]. This process can be potentiated by specific physiological stimuli or by non-specific pathological conditions, and the resulting SPM precursor is integrated into different metabolic pathways [30].

The conversion of DHA into D-series resolvins (RvD1–RvD6) requires the sequential activity of several enzymes. The transformation of DHA into RvD1 is initiated by an oxygenation reaction catalyzed by 15-lipoxygenase (15-LOX) and producing 17S-hydroperoxi-docosahexaenoic acid (17S-HpDHA), which is further subjected to (i) a peroxidase to form 17S-hydroxy-docosahexenoic acid (17S-HDHA); (ii) 5-lipoxygenase (5-LOX) producing 7-hydroperoxy-17S-HDHA (7Hp-17S-HDHA); (iii) a dehydratase leading to 7S,8S-epoxide derivative formation; (iv) finally, a hydrolase to form RvD1 and RvD2 [31]. From the pathway above, (i) 5-LOX acting on ARA or EPA with the accessory proteins 5-LOX-activating protein (FLAP) and coactosin-like protein (CLP) can produce leukotriene derivates that are conjugated with reduced glutathione (GSH) by GSH S-transferases, to form cysteinyl-containing SPMs, after the removal of the glutamate and glycine moieties by γ-glutamyl transpeptidase and dipeptidase, respectively [32]; (ii) epoxide hydrolases (EHs) are enzymes that convert epoxide containing compounds into diol products, including the soluble EH (sHE or EH2) that functions in the synthesis of resolvins shown overhead, the microsomal EH (mEH or EH1) bound to the endoplasmic reticulum catalyzing the biotransformation of xenobiotic epoxides usually formed by cytochrome P450 enzymes, which may underlie toxicity, and the recently discovered EH3 and EH4 isozymes that await further characterization [33]; (iii) a peroxidase-dependent reaction converts 7Hp-17S-HDHA in RvD5, and DHA, as a substrate for aspirin-acetylated COX-2 or P450-mixed function oxidase, can be transformed into the R isomers of AT-RvD1–AT-RvD4 [31,33,34]. Moreover, the cyclooxygenases (COXs) are heme-containing enzymes that convert arachidonic acid to prostaglandin H (PGH), which are transformed in thromboxane A_2_ and prostacyclin. There are two major isoforms of COX, namely (i) COX-1 that is a constitutive enzyme widely distributed throughout the body and believed to play a maintenance or protective role, and (ii) COX-2 that is an inducible enzyme, whose levels and activity can increase rapidly and significantly in response to stimuli, such as inflammatory mediators, thus, being associated with inflammatory processes [35]. In this context, human eosinophiles are rich in 15-LOX and convert DHA in 17-HpDHA, which can be taken up by polymorphonuclear leukocytes (PMNs) to generate RvDs [34]. Also, 17-HpDHA via 15-LOX leads to the synthesis of protectins including PD1 or its isomer PDX [36,37]. An alternative route for DHA oxygenation is catalyzed by 12-lipoxigenase (12-LOX), but also 15-LOX, to form 14-hydroxy-docosahexaenoic derivative (14-HDHA) that gives rise to the family of the maresins, namely, MaR1 and MaR2 [38]. These maresins are mainly synthesized by M2 macrophages and directly act on phagocytes [38]. The biological effects of the SPMs generated as described above are mediated by interactions with either specific receptors and intracellular effectors, or they can be re-esterified in lipid moieties [39].

The anti-inflammatory functions of SPMs are attained through binding to specialized G protein-coupled receptors (GPCRs). These include ALX/FPR2, DRV1/GPR32 and DRV2/GPR18 for resolvins, RORα and ALX/FPR2 for maresins or GPR84 and GPR120 for protectins [40,41,42]. The activation of ALX/FPR2 inhibits the phosphorylation of mitogen-activated protein kinase (MAPQ), diminishing the capacity of neutrophils and macrophages for migration and the production of proinflammatory mediators [43]. DRV1/GPR32 signaling promotes not only the macrophage polarization towards M2 and their phagocytic capacity, but also the adaptive immunological responses, whereas the DRV2/GPR18 axis regulates microglial functions [44]. Additionally, RORα controls M1 to M2 polarization in macrophages derived from infiltrated monocytes [45], while GPR38M inhibits the release of proinflammatory cytokines from macrophages and PMNs [46,47]. These receptors are expressed in several cell types and exhibit specific affinities for a given SPM [48]. Although the overexpression and knockout investigations support the receptor-mediated signaling processes to achieve anti-inflammatory effects, the signaling mechanisms are not completely understood [33].

## 4. SPMs and Regulation of Inflammatory Processes

Acute inflammation is upregulated against the aggression of microbes, lesions, and internal injuries, like those induced by surgeries [49]. Initially, the granulocytes are rapidly recruited into the infection sites, to achieve pathogen elimination through coordination with several families of pro-inflammatory cytokines and chemokines [50], an event that is unnoticed because are self-limited and naturally resolved without progressing to chronic inflammation [49]. In contrast, chronic inflammation is developed due to the inability of the host to limit the production of inflammatory factors [51]. When this condition becomes excessive and prolonged, several pathologies can appear, including rheumatoid arthritis [52] and multiple sclerosis [53], among others; consequently, it is considered an important public health problem [54]. Currently, medical research aims to describe how to control inflammation and elucidate the resolving mechanisms in order to come across the most effective treatments [55]. Inflammation resolution is an active process that is controlled by various molecular factors, such as SPMs derived from *n*-3 LCPUFAs, since they are fundamental in stopping inflammation [56]. This is achieved by controlling the magnitude and duration of resolution through (i) downregulation of the production of pro-inflammatory mediators; (ii) limitation of the traffic of leukocytes to the inflamed site; (iii) upregulation of the elimination of apoptotic cells and cellular debris by macrophages [57,58,59]; (iv) stimulation of macrophage M1 to M2 polarization; (v) enhancement in LOX activity and in the expression of SPM receptors [60,61,62]. Therefore, cells downregulate the enzymes responsible for the formation of pro-inflammatory lipids, such as prostaglandins and leukotrienes, while they upregulate those in charge of SPM production [48]. The findings acquired in experimental studies, as in human trials, show that the interruptions in the synthesis and activity of SPMs contribute to the exacerbation of inflammation [51]. Importantly, although the traditional therapeutic approaches have effectively been focused on suppressing, blocking, or inhibiting the pro-inflammatory mediators, these methodologies can provoke immunologic suppression and infections [33]. Consequently, the SPMs could be considered as an optimal therapeutic alternative, since they are not immunosuppressors compared to the prolonged use of anti-inflammatory agents, such as glucocorticoids, and they lack the toxicity of non-steroidal anti-inflammatory drugs or that of the standard procedures, e.g., chemotherapy and radiation [63,64]. Thus, experimental and clinical research on SPMs is critical, even though the quantification of SPMs has been recently questioned in terms of the analytical methods used to quantify these pro-resolving mediators in the context of their instability and their low concentrations [65]. This is particularly important regarding (i) the studies under in vitro conditions in which the storage of these compounds is crucial for the results obtained, and (ii) human studies that were not supplemented with SPMs that could lead to underestimated results [65].

## 5. Neurodegenerative Diseases

Lipids represent up to 50% of the brain dry weight and they are the main structural components of the cellular membranes, which were found to be de-regulated in neurodegenerative diseases [24]. These neurological disorders are characterized by a chronic inflammatory process, wherein the resolution mechanism is altered [66].

### 5.1. Alzheimer Disease (AD)

AD is considered the most common type of dementia that is characterized by the accumulation of the β-amyloid protein in the human brain and by the formation of the neurofibrillary tangles as the main histopathologic markers [67]. When brain β-amyloid levels are high, the innate immune cells are activated, thus, triggering the pro-inflammatory signaling pathways that may alter the neuronal functions [68]. In this respect, it has been shown that SPMs induce a significant increase in the resolution of the inflammation routes in AD, strongly suggesting that these mediators may be promising therapeutic strategies [69]. In a recent study, the analysis of the lipidome of the cerebrospinal fluid revealed that the SPMs were diminished in AD, in correlation with the subjective cognitive impairment and with the significant enhancement in the levels of pro-inflammatory mediators [70]. Mizwicki et al. reported that the microglial phagocytosis of β-amyloid was enhanced by RvD1, an effect that was concentration-dependent and promoted cell survival [71]. In agreement with these findings, in vitro studies in neuronal models indicate that cellular survival improvement by RvD1 [26] is accompanied by a diminution in β-amyloid production [72], while those in vivo showed an attenuation of cognitive decline, reduction in neuroinflammation, and amendment of memory [73,74,75].

### 5.2. Parkinson Disease (PD)

PD is the second most common degenerative disease of the central nervous system [76]. This disease is characterized by motor and non-motor symptoms, including tremor at rest, rigidity, bradykinesia, postural instability, constipation, and depression [77]. The main pathological features of PD are the progressive degeneration of dopaminergic neurons located in the dense part of the substantia nigra, leading to a diminution in dopamine concentrations in the striatum [78] and an accumulation of the protein α-synuclein, which forms Lewy bodies [79]. Lewy bodies are composed of protein aggregates of α-synuclein with a minor contribution of neurofilament protein, ubiquitin, and α-B-crystallin, leading to mitochondrial dysfunction triggering oxidative stress, further protein misfolding and stimulating the fibrillar pathway, events that determine neurodegeneration [77,78,79]. Although the knowledge of the pathogeny of PD has experienced important progress, few advances have been achieved on the effects of the stimulation of inflammation resolution [80]. According to Xu et al., RvD1 inhibits the synthesis of inflammatory mediators in microglia and the expression of tumor necrosis factor-α (TNF-α), interleukin-1β (IL-1β), and inducible nitric oxide synthase (NOS) [79], attenuating the microglial expression of NF-*κ*B and activating protein-1 (AP-1) and MAPK phosphorylation [81]. The administration of different concentrations of RvD1 in an experimental model revealed the decrease in PD progression due to the inhibition of inflammation [82]. Tian et al. studied the effect of an intrathecal injection of RvD2 in an animal model, showing that the treatment prevented the development of behavioral defects and the activation of the toll-like receptor 4 (TLR4)/NF-κB signaling pathway; therefore, there was a decrease in pro-inflammatory mediators and in the production of reactive oxygen species (ROS) [83]. This effect is not only exerted by the resolvin family; neuroprotectin D1 (NPD1), which has been shown to promote survival and the preservation of the dendritic tree in rat dopaminergic neurons in vitro, also plays a role [84]. It is important to mention that AD and PD would not be the only beneficiaries of SPM actions, since there is evidence that other alterations in the central nervous system can benefit from them. These include autoimmune encephalomyelitis [85], multiple sclerosis [86] and amyotrophic lateral sclerosis [87]. All these findings lay the foundation for understanding how the survival of neurons can be improved and how to reduce neuroinflammation in order to avoid neurodegeneration.

## 6. Respiratory Diseases

SPMs specialized in airway inflammatory response have been used for the treatment of respiratory disease [88]. In this regard, respiratory diseases include a wide range of pathologies with different clinical manifestations, affecting the normal airways and lung function. An increase in the inflammatory response is considered a characteristic point of these diseases, being also a critical factor for their progression [88]. In this context, asthma is a chronic inflammatory disease which has no cure. It is characterized by bronchial hyperresponsiveness, airflow obstruction, and airway inflammation [89]. Glucocorticoids have become the first choice for the treatment of asthma due to their anti-inflammatory effects; however, long-term use may cause side effects. Therefore, there is a need to develop alternative strategies [90]. Several studies have demonstrated that the SPMs derived from DHA and its precursors are deregulated in asthmatics [91,92]. In a mouse model of induced asthma, MaR2 exerted anti-inflammatory effects through the inhibition of oxidative stress, inflammasome NLRP3 activation, and type Th2 immune response [93]. Ou et al. used a rat model to show that MaR1 notably suppressed the activation of the NF-*κ*B signaling pathway as well as those of COX-2 and ICAM-1 [94]. Furthermore, in type 2 innate lymphoid cells (ILC2), exogenous MaR1 diminished the pulmonary inflammation and IL-5 and IL-13 expression, augmenting the de novo generation of regulatory T cells (Tregs) [95]. Consequently, alterations in the synthesis of SPMs and the persistence of inflammation could be important mechanisms to explain the chronic nature of the inflammatory process, pointing to SPMs derived from DHA as an effective strategy for asthma [96]. Furthermore, the global outbreak of coronavirus disease 2019 (COVID-19), which originated in Wuhan, China, has claimed millions of lives worldwide; therefore it has been a disease of quite interest recently [97]. Some studies have described an imbalance in the SPMs as a defining characteristic of the severity of COVID-19 [98,99]. One of the main consequences and reasons for concern is pneumonia, which is not only caused by SARS-CoV-2, but also by other pathogens, including rhinovirus, influenza A or B virus, respiratory syncytial virus or adenovirus, physical and chemical factors, immune disorders, allergies, and medications [100]. Pneumonia consists of the inflammation of the terminal airways, the alveoli, and the interstice of the lungs [101]. Studies with experimental animals demonstrated that expression of RvD1 is able to significantly diminish pneumonia caused by *P. aeruginosa* [102], since bacterial growth, leukocyte infiltration, and damage to lung tissue are decreased [103]. It has been reported that inflammation and lung injury were persistent in pneumonia induced by *Haemophilus influenzae* in rats [104]. However, the exogenous administration of RvD1 reduced neutrophil recruitment, increased macrophage entry, stimulated macrophage M1 to M2 polarization, and lowered IL-6 and TNF-α expression [104]. Moreover, influenza virus decreased lung PD1 levels during severe infections, yet the exogenous treatment improved rat survival in the infected animals even at later stages of the disease [105]. It was also shown that PD1 is a potential molecule to prevent the spread of H5N1 virus. In this regard, Ramon et al. communicated that the product of DHA, 17-HpDHA, could promote the significant increase in the levels of serum antibodies, and enhance the number of antibody secretory cells in the bone marrow of rats [106].

## 7. Metabolic Syndrome

Obesity and metabolic disorders are important public health problems around the world [107]. As obesity rises, the immunological profile of the adipose tissue changes, going into a chronic inflammatory state of low grade, which gradually becomes systemic and develops insulin resistance and metabolic disease [108,109]. In this context, Titos et al. studied the inflamed human adipose tissue and observed that RvD1 treatment enhanced the MAPK activity, concomitantly with a diminution in signal transducer and activator of transcription 1 (STAT1) functioning and related inflammatory gene expression, without altering the anti-inflammatory effects of IL-10 [110]. In addition, treatment with RvD1 and MaR1 was found to polarize macrophages towards a phenotype similar to M2, decreasing the levels of pro-inflammatory markers in adipose tissue of obese mice [111,112]. An in vivo study revealed a decrease in obese mice adipose tissue related to RvD1, RvD2, and PD1 amounts, compared to that of lean mice, identifying RvD1 and RvD2 as the main SPMs that reduce inflammatory processes in adipose tissue [113].

Obesity diminishes the levels of PD1 of the intermediates in the synthesis of resolvins and protectins (17-HDHA) as well as maresins (14-HDHA), in the adipose tissue of obese mice induced by diet or genetically [113,114,115]. Mice with leptin receptor deficiency given RvD1 exhibit an improvement in glucose tolerance and insulin sensitivity, along with a reduction in pro-inflammatory gene expression and the inflammatory macrophage formation [116]. Furthermore, treatment with MaR1 reversed the effect of the pro-inflammatory cytokine TNF-α and induced the phosphorylation of protein kinase B (Akt) in subcutaneous adipose tissue of obese patients, and also improved glucose homeostasis in obese mice [117]. This latter beneficial effect of MaR1 was suggested to be mediated by fibroblast growth factor-21 (FGF21) [118], a peptide hormone mainly synthesized in the liver that contributes to the regulation of glucose and lipid metabolism and energy homeostasis [119]. Consequently, DHA-derived SPMs protect against adipose tissue inflammation and insulin resistance brought on by obesity; therefore, they could be new therapeutic options for the therapy of metabolic syndrome.

## 8. Cardiovascular Diseases

Atherosclerosis is a disease associated with the inflammation and dysfunction of lipid metabolism in the arteries, driven by lipid imbalance of the pro-inflammatory and resolution mechanisms [120]. It has been identified that SPMs, especially RvD1, are decreased in vulnerable regions, histologically defined as human carotid atherosclerotic plaques [121]. Rats subjected to the administration of RvD2 prevented atheroprogression by suppressing endothelial cell necrosis and collagen fibrous plaque formation and inhibited the secretion of mature IL-1β by bone marrow-derived macrophages challenged with LPS + ATP [122]. These findings are of great interest because IL-1β can induce a *cytokine storm* in the host [123]. Moreover, MaR1 decreased cell-to-cell adhesion of monocytes and vascular cells, elicited attenuation of NF-κB activation by TNF-α in endothelial cells, and lowered the levels of pro-inflammatory cytokines and chemokines [124]. In addition, it has been observed that RvD1 regulates human PMN recruitment and SPM synthesis [125], PDs and MaRs can play an effective role in the pathogenesis associated with worsening cardiometabolic status [126].

Ischemic heart disease is the main cause of disability and death in the whole word and is the result of an insufficient supply of blood and oxygen to the heart [127]. It has been reported that rats subjected to RvD1 at the beginning of ischemia in vivo decreased infarct size by reducing the mechanism involving phosphoinositide 3-kinase (PI3K)/Akt [128]. Under these conditions, RvD1 limits neutrophil recruitment in the spleen and left ventricle, augments inflammation resolution, and increases the expression of resolving M2 macrophage markers after myocardial infarction [129]. Lastly, Gilbert et al. administered RvD1 to rats subjected to ischemia/reperfusion, showing attenuation of the symptoms of myocardial depression and the size of the infarct [130].

It is important to point out that atherosclerosis and ischemic heart disease are not the only pathologies that would benefit from DHA-derived SPMs, since Pope et al. reported the attenuation of the formation and progression of aneurysms in murine models, through polarization of the aortic wall macrophages towards a reparative M2 phenotype [131]. Also, early platelet–neutrophil interactions at sites of injury or thrombosis lead to MaR1 biosynthesis that stimulates the onset of resolution [132], an SPM that improves the hemostatic function of human platelets and suppress their inflammatory functions [133]. The protecting actions of SPMs are not limited to the heart, having been described in the context of ischemia in brain, kidneys, and liver [134].

## 9. Liver Diseases

Chronic liver disease is more often associated with the ailment known as non-alcoholic fatty liver disease (NAFLD), which is characterized by a process of continuous inflammation [135]. NAFLD involves two phases, namely non-alcoholic fatty liver (NAFL) and non-alcoholic steatohepatitis (NASH), a condition that includes degrees of fibrosis, cirrhosis, and hepatocellular carcinoma [136]. In this scenario, SPMs could be a treatment option for the active promotion of the cessation of inflammation [64]. Accordingly, in mice subjected to a high-fat diet, MaR1 improves hepatic steatosis by inhibiting endoplasmic reticulum (ER) stress and lipogenic enzymes and inducing autophagia via AMP-activated protein kinase (AMPK) [137,138,139]. In the model of HepG2 cells, RvD1 also decreases ER stress and the dependent caspase-3/apoptosis activation, with diminution of triacylglycerol accumulation [140], findings that were reproduced by PDX [141].

Experimental studies have revealed that NASH can also be alleviated by SPM administration, considering that (i) MaR1 exerted liver protection by activating the M2 polarization of Kupffer cells [142]; (ii) RvD1 similarly stimulated liver macrophage M1 to M2 phenotype polarity, in addition to an anti-steatosis effect and macrophage infiltration arrest [143]; (iii) the establishment of negative correlations between the serum levels of MaR1 and body mass index, waist circumference, alanine transaminase, gamma-glutamyl transpeptidase, uric acid, triglyceride, and fasting blood glucose [144]; (iv) MaR1 improves parameters related to hepatic fibrosis, concomitantly with an improvement in hepatocyte proliferation and diminution of oxidative stress and inflammation [145], supporting SPMs as potential therapeutic agents for NAFLD, NASH, and other liver pathologies.

Our research group have suggested that combining *n*-3 PUFAs and other protecting agents may result in better responses than monotherapies concerning NAFLD [146]. This strategy underlies (i) protective agents exerting their actions through different or similar mechanisms to achieve synergistic or additive results to control the damage more efficiently, and (ii) the minimization of possible side effects due to the utilization of lower dosages than the monotherapies and shorter administration periods [147]. Data reported using the high-fat diet protocol for 12 weeks revealed that the concomitant EPA plus hydroxytyrosol (HT) supplementation synergistically diminished the steatosis score over individual treatments, increasing the liver levels of EPA, DHA, RvD1/2, and RvE1/2, and attenuating inflammation [148]. More interestingly, DHA and HT co-administration confronting a high-fat diet fully precluded liver steatosis and the pro-inflammatory state [149] compared to the EPA plus HT protocol [147], a contention that may be related to the higher chemical reactivity of DHA generating active derivatives affording more beneficial effects than EPA [14]. Co-supplementation of the hormetic hormone L-3,3′,5-triiodothyronine (T_3_) with either (i) methylprednisolone, to preserve liver tissue regeneration post-hepatectomy [150], or (ii) fish oil, to suppress ischemia-reperfusion inflammatory liver injury [151], have been also suggested.

## 10. Other Pathologies

SPM derivatives of DHA are linked to eye health. It has been described as a disturbance in the homeostasis of mucin secretion produced by conjunctival goblet cells, in a variety of ocular surface diseases, such as allergic conjunctivitis and dry eye disease [152]. Alterations in the quantity, structure, or hydration of mucin are detrimental to the clarity of the cornea and, therefore, for vision [153]. Different studies have demonstrated in a rat model that RvD1, MaR1, and MaR2 modulate the function of the conjunctival goblet cells to produce mucin and, therefore, maintain homeostasis of the ocular surface and lachrymal film [154,155]. An effect on the aging of the retina has also been seen, as evidenced by a recent study of Trotta et al., who observed in aged rats that the levels of RvD1 in the retina were diminished [156].

A relationship has been also observed with some (i) dental pathologies, such as inflammatory periodontal disease, which progresses rapidly and causes destruction of the supporting tissues of the teeth [157], with MaR1 improving the phagocytosis and destruction of periodontal pathogens [28]; likewise, orthodontic treatment produces a mechanical force that triggers an acute inflammatory process driven by cells and immune mediators [158], where, in the acute phase of inflammation, exogenous RvD1 favors resolution, whereas, in the prolonged phase, it suppresses osteoclast genesis [159]. (ii) Psoriasis would also be a disease that would benefit from DHA-derived SPMs, since PD1 decreases the symptoms of the disease including desquamation and erythema with a reduction in pro-inflammatory cytokine and chemokine formation, and improving the thickness of the skin [160]. Finally, (iii) SPMs derived from DHA are also associated with male infertility [161], arthritis [162], cystitis [163], and even postmenopausal osteoporosis [164], as well as the other clinical applications mentioned above (Figure 3).

## 11. Conclusions

It has been shown that inflammation is a pathophysiological trait which plays a crucial role in the pathogenesis of various diseases. The identification of biochemical pathways that actively mediate the resolution of inflammation offers new treatment opportunities and monitoring of progression and disease prognosis. Growing evidence indicates that DHA-derived SPMs have important anti-inflammatory and pro-resolving properties, so they have been considered as possible therapeutic strategies in various pathological conditions. To date, several experimental studies have evaluated the effectiveness of D-series resolvins, protectins and maresins, either alone or in combination with other protective agents. There is an urgent need to further investigate the therapeutic role of these lipid mediators in the clinical setting, in order to accurately identify molecular and cellular resolution pathways in inflammatory pathologies and to provide therapies that foreshadow effective future clinical applications.

## Figures and Tables

**Figure 1 nutrients-15-03317-f001:**
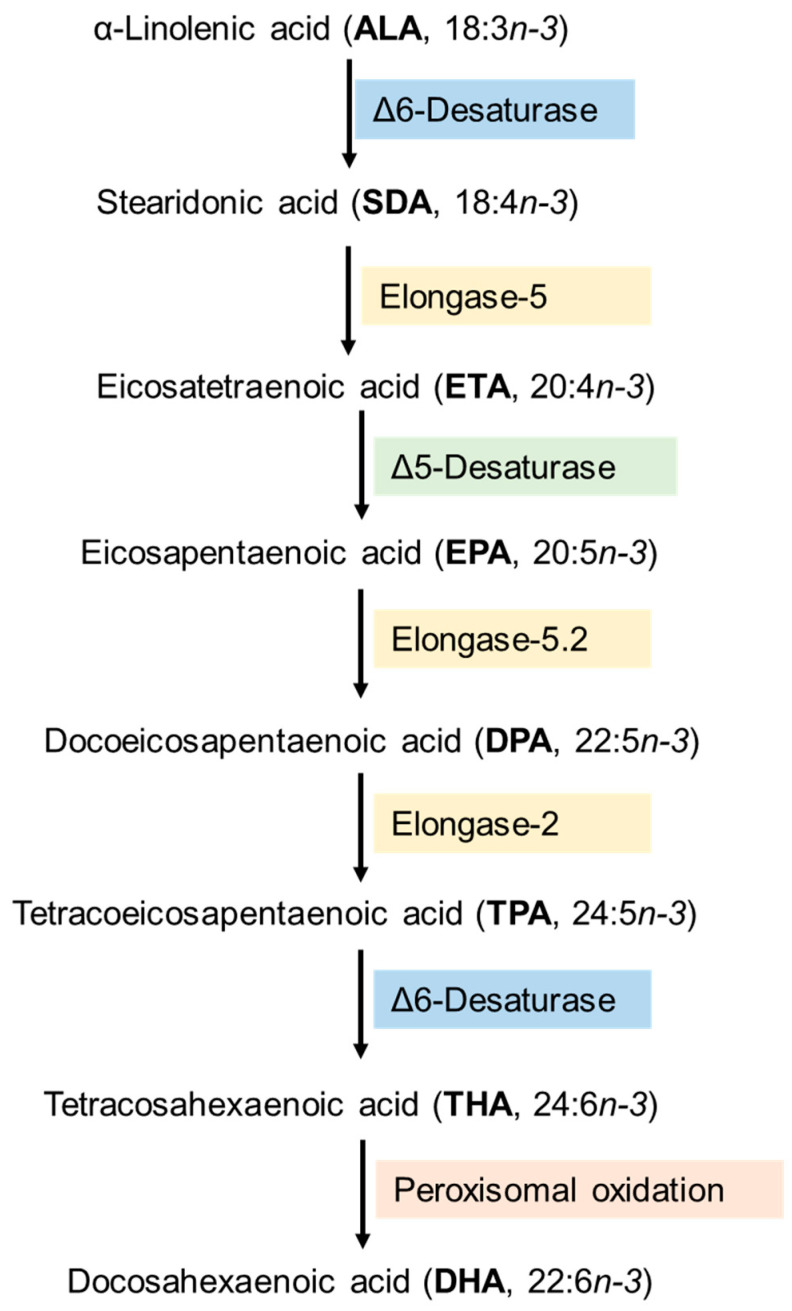
Metabolic pathway of the synthesis of *n*-3 polyunsaturated fatty acids from α-Linolenic acid.

**Figure 2 nutrients-15-03317-f002:**
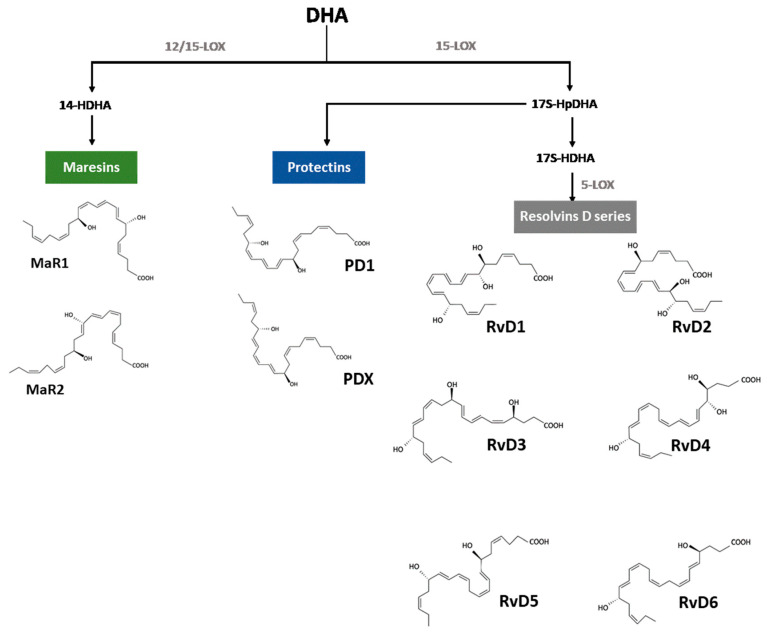
Structure and biosynthesis of the SPMs: maresins, protectins, and D-series resolvins, which are metabolites derived from DHA.

**Figure 3 nutrients-15-03317-f003:**
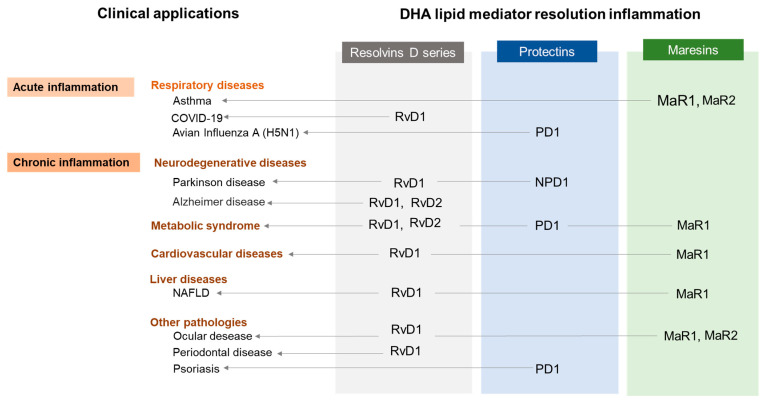
DHA lipid mediator resolution inflammation and clinical applications.

## Data Availability

Not applicable.

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
