# Peer review of "Potential Clinical Applications of Pro-Resolving Lipids Mediators from Docosahexaenoic Acid"

_nutrients, 2023, doi:10.3390/nu15153317_

Round 1
Reviewer 1 Report
The paper is well-written and a comprehensive review of DHA-derivated SPMs effects in significant clinical pathologies has been described.
A few minor questions and corrections must be pointed out.
- in the abstract "Growing evidence suggests that these metabolites would have important pro-resolving properties" should include inflammatory pro-resolving properties.
- on page 2 line 61:
"The SPMs are formed through enzymatic or non-enzymatic pathways. " What are the non-enzymatic SPMs? And as non-enzymatic oxidation of DHA or SPMs was not discussed in the review this sentence should be modified.
- Some reference numbers on the list are duplicated, (113, 114, 141-146...)
- Last year a group of researchers published a review questioning the quantification of SPMs (10.3389/fphar.2022.838782), with that in consideration I would suggest the authors add as part of the discussion when they present data from human studies that were not supplemented with SPM some information regarding the quantification.
Also worth mentioning is that DHA and the DHA-derivated SPMs are highly susceptible to peroxidation and that also limits the results obtained. storage of these compounds is crucial for the results obtained.
Author Response
Reviewer #1
Thank you for reviewing our manuscript entitled “Potential Clinical Applications of Pro-Resolving Lipids Mediators from Docosahexaenoic Acid”, We have prepared a revised version of the manuscript according to your comments.
Comments and Suggestions for Authors
The paper is well-written and a comprehensive review of DHA-derivated SPMs effects in significant clinical pathologies has been described.
A few minor questions and corrections must be pointed out.
1.- in the abstract "Growing evidence suggests that these metabolites would have important pro-resolving properties" should include inflammatory pro-resolving properties.
Answer: in the Abstract, “pro-resolving properties” was changed to “inflammatory pro-resolving properties”
2.- on page 2 line 61:
"The SPMs are formed through enzymatic or non-enzymatic pathways. " What are the non-enzymatic SPMs? And as non-enzymatic oxidation of DHA or SPMs was not discussed in the review this sentence should be modified.
Answers:
2.1.- The non-enzymatic SPMs were included in the following paragraph in lines 59 to 62:
“The non-enzymatic synthesis is mediated by free radical related peroxidation processes [21], leading to the formation of different pro-resolving derivates of EPA including hydroxyeicosapentaenoic acids (HEPEs), such as 11-HEPE and 18-HEPE, among others [21,22].”
New references 21 and 22 were added to the list, and the rest were re-umbered in text and list.
2.2.- Non-enzymatic SPMs were not discussed, as already indicated in lines 51-53:
“This review addresses only the resolvins, protectins and maresins, due to the solid information on these mediators available at the present time.”
3.- Some reference numbers on the list are duplicated, (113, 114, 141-146...)
Answer: these references (113 and 114, 141 to 146) are different citations, what was duplicated were the reference numbers, which were eliminated.
4.- Last year a group of researchers published a review questioning the quantification of SPMs (10.3389/fphar.2022.838782), with that in consideration I would suggest the authors add as part of the discussion when they present data from human studies that were not supplemented with SPM some information regarding the quantification.
Answer: this comment and that below (5) are answered in the following paragraph added to lines 160-166:
“Thus, experimental and clinical research on SPMs is critical, even though the quantification of SPMs has been recently questioned in terms of the analytical methods used to quantify these pro-resolving mediators in the context of their instability and their low concentrations [64]. This is particularly important regarding (i) studies under in vitro conditions in which the storage of these compounds is crucial for the results obtained; and (ii) human studies that were not supplemented with SPM that could lead to underestimated results [64].
New reference 64 was added to the list of references, and the rest were re-numbered in text and list.
5.- Also worth mentioning is that DHA and the DHA-derivated SPMs are highly susceptible to perox idation and that also limits the results obtained. storage of these compounds is crucial for the results obtained.
Answer: this comment was answered above (comment 4)
Reviewer 2 Report
The review submitted by Beyer, M.P.; Videla, L.A.; Farías, C.; and Valenzuela, R., gives a wide description on the clinical applications of DHA-derived specialized pro-resolving mediators (SPMs). SPMs are endogenous small molecules that counter-regulate pro-inflammatory signals and allow inflamed tissues to return to homeostasis once the need for the inflammatory response is over – without immunosuppression. Therefore, the SPMs including those produced from other related PUFAs have garnered great interest over the last few decades, and present a novel therapeutic approach to treating inflammation in a wide range of pathologies.
This review presents itself well overall and covers a large breadth of material. However, I have some suggestions and minor comments listed below.
Lines 53-54: reads "... the specific pro-resolving mediators...". It should read, "... the specialized pro-resolving mediators...".
Lines 61-68: The authors should also consider discussing the role of hydrolase enzymes in the biosynthesis of SPMs.
Fig. 2: The MaR1 and MaR2 are incorrectly named – the MaR1 chemical structure is 7R,14S-dihydroxylated SPM whereas MaR2 is 13R,14S-vicinal diol SPM. If the authors have access to a molecular editing software such as ChemDraw, they should redraw the structures so that they all look uniform in appearance and size and thus more appealing to the reader.
Along these lines, even though the authors don't discuss the cysteinyl-containing SPMs in the review, they could show and discuss their biosynthesis from DHA.
Line 77: reads, "resolvins D series.." should read, " D-series resolvins"
Line 102: reads, "macrophages M2.." should read, "M2 macrophages.."
References 124-125 appear to be out of order in which they're cited.
References 135-137 appear to be out of order in which they're cited.
Lines 239: change drug to molecule or mediator, etc.
Lines 240-241: 17-HpDHA is a product of DHA, not a precursor.
Lines 378-9: reads, "series D resolvins.." should read, " D-series resolvins"
The overall English quality is great but I'd advise that the authors revise the text for sentence structure and minor grammatical errors.
Line 56: reads, "solid information...". Consider switching to, "definitive information.." or "experimental or research data.."
Lines 88-9: reads, "17S-hydroperoxi-docosahexaenoic acid.." should read, "17S-hydroperoxy-docosahexaenoic acid.."
Line 107: reads, "G protein-couple receptors" should read, "G protein-coupled receptors"
Line 171: reads, "In agreement these findings,.." should read, "In agreement with these findings,.."
Line 199: Revise the whole sentence structure.
Line 203: Revise the whole sentence.
Line 253: Revise the whole sentence.
Line 269: Revise the whole sentence.
Line 287: Revise the sentence... world not word.
Line 300: Revise the whole sentence.
Line 313: Add parenthesis to ER
Line 368: Revise, incomplete sentence.
Author Response
Reviewer #2
Thank you for reviewing our manuscript entitled “Potential Clinical Applications of Pro-Resolving Lipids Mediators from Docosahexaenoic Acid”, We have prepared a revised version of the manuscript according to your comments.
Comments and Suggestions for Authors
The review submitted by Beyer, M.P.; Videla, L.A.; Farías, C.; and Valenzuela, R., gives a wide description on the clinical applications of DHA-derived specialized pro-resolving mediators (SPMs). SPMs are endogenous small molecules that counter-regulate pro-inflammatory signals and allow inflamed tissues to return to homeostasis once the need for the inflammatory response is over – without immunosuppression. Therefore, the SPMs including those produced from other related PUFAs have garnered great interest over the last few decades, and present a novel therapeutic approach to treating inflammation in a wide range of pathologies.
This review presents itself well overall and covers a large breadth of material. However, I have some suggestions and minor comments listed below.
1.- Lines 53-54: reads "... the specific pro-resolving mediators...". It should read, "... the specialized pro-resolving mediators...".
Answer: “specific” was changed to “specialized” in line 17, as well as in line 10 (Abstract).
2.- Lines 61-68: The authors should also consider discussing the role of hydrolase enzymes in the biosynthesis of SPMs.
Answer: Thank you very much for this comment, we added a new paragraph in lines 96-101:
“(ii) epoxide hydrolases (EHs) are enzymes that convert epoxide containing compounds into diol products, including the soluble EH (sHE or EH2) that functions in the synthesis of resolvins shown overhead, the microsomal EH (mEH or EH1) bound to the endoplasmic reticulum catalyzing the biotransformation of-xenobiotic epoxides usually formed by cytochrome P450 enzymes, which may underlie toxicity, and the recently discovered EH3 and EH4 isozymes that await further characterization [33].”
New reference 33 was added to the list of references, and the rest were re-numbered in text and list.
3.- Fig. 2: The MaR1 and MaR2 are incorrectly named – the MaR1 chemical structure is 7R,14S-dihydroxylated SPM whereas MaR2 is 13R,14S-vicinal diol SPM. If the authors have access to molecular editing software such as ChemDraw, they should redraw the strctures so that they all look uniform in appearance and size and thus more appealing to the reader.
Answer: Thanks for this comment, we improved Figure 2.
4.- Along these lines, even though the authors don't discuss the cysteinyl-containing SPMs in the review, they could show and discuss their biosynthesis from DHA.
Answer: As requested by this referee, the following paragraph was introduced in lines 93-96:
“5-LOX acting on ARA or EPA with the accessory proteins 5-LOX-activating protein (FLAP) and coactosin-like protein (CLP) can produce leukotriene derivates that are conjugated with reduced glutathione (GSH) by GSH-S-transferases, to form cysteinyl-containing SPMs after the removal of the glutamate and glycine moieties by γ-glutamyl transpeptidase and dipeptidase, respectively [32];”
New reference 32 was added to the list, and the rest of them were re-numbered in text and
5.- Line 77: reads, "resolvins D series.." should read, " D-series resolvins"
Answer: “resolvins D series” was changed to “D-series resolvins” in line m78, as well as in lines 50, 88 and 398.
6.- Line 102: reads, "macrophages M2.." should read, "M2 macrophages"
Answer: “macrophages M2” was transformed into “M2 macrophages” in line 112.
7.- References 124-125 appear to be out of order in which they're cited.
Answer: due to the addition of new references, ref. 124-125 became ref. 131-132 (lines 318-320) in the revised version. Both referred to the effects of MaR1 on platelet responses, and they are in agreement with the text.
8.- References 135-137 appear to be out of order in which they're cited.
Answer: due to the addition of new references, ref. 135-137 became ref. 140-142 (lines 335-338) in the revised version. They referred to the effects of SPMs on liver steatosis or steatohepatitis in different models, and they agree with the text.
9.- Lines 239: change drug to molecule or mediator, etc.
Answer: “drug” was converted into “molecule” in line 259.
10.- Lines 240-241: 17-HpDHA is a product of DHA, not a precursor.
Answer: “precursor” was changed for “product” in line 260.
11.- Lines 378-9: reads, "series D resolvins.." should read, " D-series resolvins"
Answer: this comment was already answered in question 5 of this referee.
Reviewer 3 Report
Reviewer comments
The article contains errors in the construction of sentences, that in some instances prevent the reader from grasping the meaning behind some paragraphs. I suggest revising the English grammar. I have indicated some mistakes but not all of them.
Title
The title does not reflect what is the submission about. For instance, I could not find any description or discussion of the referred “potential molecular applications”. In addition, the title of the article implies that “targeting lipid mediators from DHA” is an important component of the submission. However, I cannot find anything in this specific context. I suggest rephrasing the title. For instance: “Potential clinical applications of pro-resolving lipids mediators from docosahexaenoic acid”.
Abstract
The section is very repetitive and contains some sentence construction problems. I have removed the repetitive statements and superfluous information and reduced this section significantly.
I suggest the following abstract:
“Docosahexaenoic acid (C22:6n-3, DHA) is the precursor of pro-resolving lipid mediators (SPMs), such as resolvins, protectins and maresins families which have been considered therapeutic bioactive compounds for human health. Growing evidence regards DHA and SPMs as beneficial strategies in the amelioration, regulation, and duration of inflammatory processes through different biological actions. The present review discusses the reported therapeutic benefits of DHA and SPMs on various diseases and their potential clinical applications.”
Instead of the submitted abstract:
“Docosahexaenoic acid (C22:6n-3, DHA) is a long-chain polyunsaturated fatty acid that is considered an important bioactive compound for human health, which is a precursor of pro-resolving lipid mediators (SPMs) that include the resolvins, protectins and maresins families. Growing evidence suggests that these metabolites would have important pro-resolving properties, so they have been considered as possible therapeutic strategies in several pathological contexts, by being essential for the cessation of inflammation as they can regulate its magnitude and duration through different biological actions. This literature review is focused on providing a broad description of the current evidence on the different clinical applications of DHA and DHA-derived SPMs, with the aim of clarifying the benefits they would have on various diseases, considering the urgent need to further investigate the therapeutic roles of these SPMs, in order to more precisely identify their future clinical application.”
Introduction
Line 29. “The Food and Agriculture Organization (FAO-2010) advises…” instead of “The Food and Agriculture Organization (FAO-2010) of the United Nations advises…”
Lines 45. Comparative adjectives such as “slower” cannot close a sentence. It needs an additional object to establish the comparison. For instance, “X is slower than Y.”
Lines 53-54. “specific pro-resolving mediators” Why in italics?
Lines 55-57. “This review addresses only the resolvins, protectins and maresins, due to the solid information on these mediators available at the present time.” The abstract section states that DHA, resolvins, protectins and maresins are considered. However, according to the quoted statement DHA is not discussed. I suggest amending the abstract section accordingly. Also, I suggest rephrasing and improving the scientific style of the statement in question.
Line 62. The authors state “cyclooxygenases”, however the role of these enzymes on the production of SPMs is not clear throughout the manuscript. In addition, the only instance where these enzymes are mentioned is to indicate just COX-2 (lines 94 and 216). It seems that other COXs have not involvement in SPMs production.
Methods
Line 72. “2010 onwards, including” instead of “2010 on including”
Synthesis and metabolism of DHA-derived lipid mediators
Line 106. “The anti-inflammatory functions of SPMs favoring resolution are attained…” Stating anti-inflammatory function favouring resolution is redundant. I suggest “The anti-inflammatory functions are attained…”
SPMS and regulation of inflammatory processes
Line 127. “In contrast, chronic inflammation…” instead of “Contrarily to acute inflammation, chronic inflammation…”
Lines 145-148. “All of the above is of particular interest, considering that the traditional therapeutic approaches they have focused to suppress, block or inhibit the pro-inflammatory mediators, and, although several of them are effective, they can provoke immunologic suppression and infections [30].” What is “All of the above”? Who are “they” or them? The quoted statement is difficult to digest. Could you please rephrase it in a comprehensive, clear, grammatically well-constructed and more scientific style.
Line 171. “In agreement these findings…” grammatically incorrect.
Line 196. “tthrefore,…” delete extra letter
Line 199. “…reservation of the dendritic tree” What does the quoted statement mean?
Respiratory diseases
It is incorrect to start this section by describing one of the several existing respiratory diseases. I suggest starting by explaining that SPMs have been used for the treatment of respiratory diseases and after that the authors can elaborate on the different respiratory diseases. Also, the authors can start by giving a more general statement on respiratory diseases as in section 7 for metabolic syndrome.
Lines 233-234. “It has been reported that inflammation and lung injury were persistent in pneumonia induced by Haemophilus influenzae in rats [98]” instead of “In pneumonia induced by Haemophilus influenzae in rats, Croasdell et al. found that inflammation and lung injury were persistent [98]”
Metabolic syndrome
Lines 253-253. “. In addition to the above data” What data are you referring to?
Cardiovascular diseases
Line 281. According to the Oxford and Cambridge dictionaries, the expression “on the other hand” should be preceded by 'on the one hand'. Consequently, the authors should rephrase the whole paragraph.
Liver diseases
Lines 312-314. “… endoplasmic reticulum (ER) stress and lipogenic … decreases ER stress…” instead of “… endoplasmic reticulum ER stress and lipogenic … decreases ER stress…”
Lines 327-328. “Research by our group evaluated the proposal that combined protocols involving n-3 PUFAs and other protecting agents may result in better responses than monotherapies…” This is an odd statement. I suggest rephrasing it. For example: “Our research group have suggested that combining n-3 PUFAs and other protecting agents may result in better responses than monotherapies…”
Lines 340-344. I suggest rephrasing as “Co-supplementation of the hormetic hormone L-3,3′ ,5-triiodothyronine (T3) with either (i) methylprednisolone, to preserve liver tissue regeneration post-hepatectomy [145]; or (ii) fish oil, to suppress ischemia-reperfusion inflammatory liver injury [146] have been also suggested .” instead of “Other instances supporting the above proposition is the co-supplementation of the hormetic hormone L-3,3′ ,5-triiodothyronine (T3) with either (i) methylprednisolone, to preserve liver tissue regeneration post-hepatectomy [145]; or (ii) fish oil, to suppress ischemia-reperfusion inflammatory liver injury [146].”
Line 356. As mentioned above, the Oxford and Cambridge dictionaries state that the expression “on the other hand” should be preceded by 'on the one hand'. Consequently, the authors should rephrase the whole paragraph.
References
Remove the double reference numbering. For example, 113, 114, 142-146
References 5, 38, 75, 87, 92, 133 should have space between year and volume. Check all the references.
Some references are given by capitalizing every first letter (e.g., 1, 3, 27, 29, etc) of the title, while other references only capitalized the first letter of the title (e.g. 2, 15, 16, etc). Try to be consistent.
Very bad quality. Some sentences are too long and contain superfluous and repetitive information. The authors should improve the English level and use a more scientific style.
Author Response
Reviewer #3
Thank you for reviewing our manuscript entitled “Potential Clinical Applications of Pro-Resolving Lipids Mediators from Docosahexaenoic Acid”, We have prepared a revised version of the manuscript according to your comments.
Comments and Suggestions for Authors
The article contains errors in the construction of sentences, that in some instances prevent the reader from grasping the meaning behind some paragraphs. I suggest revising the English grammar. I have indicated some mistakes but not all of them.
1.- Title
The title does not reflect what is the submission about. For instance, I could not find any description or discussion of the referred “potential molecular applications”. In addition, the title of the article implies that “targeting lipid mediators from DHA” is an important component of the submission. However, I cannot find anything in this specific context. I suggest rephrasing the title. For instance: “Potential clinical applications of pro-resolving lipids mediators from docosahexaenoic acid”.
Answer: The title was rephrased as indicated by the reviewer (lines 2-3).
2.- Abstract
The section is very repetitive and contains some sentence construction problems. I have removed the repetitive statements and superfluous information and reduced this section significantly.
I suggest the following abstract:
“Docosahexaenoic acid (C22:6n-3, DHA) is the precursor of pro-resolving lipid mediators (SPMs), such as resolvins, protectins and maresins families which have been considered therapeutic bioactive compounds for human health. Growing evidence regards DHA and SPMs as beneficial strategies in the amelioration, regulation, and duration of inflammatory processes through different biological actions. The present review discusses the reported therapeutic benefits of DHA and SPMs on various diseases and their potential clinical applications.”
Instead of the submitted abstract:
“Docosahexaenoic acid (C22:6n-3, DHA) is a long-chain polyunsaturated fatty acid that is considered an important bioactive compound for human health, which is a precursor of pro-resolving lipid mediators (SPMs) that include the resolvins, protectins and maresins families. Growing evidence suggests that these metabolites would have important pro-resolving properties, so they have been considered as possible therapeutic strategies in several pathological contexts, by being essential for the cessation of inflammation as they can regulate its magnitude and duration through different biological actions. This literature review is focused on providing a broad description of the current evidence on the different clinical applications of DHA and DHA-derived SPMs, with the aim of clarifying the benefits they would have on various diseases, considering the urgent need to further investigate the therapeutic roles of these SPMs, in order to more precisely identify their future clinical application.”
Answer: The abstract was re-written as suggested by the reviewer (lines 10-15).
Introduction
3.- Line 29. “The Food and Agriculture Organization (FAO-2010) advises…” instead of “The Food and Agriculture Organization (FAO-2010) of the United Nations advises…
Answer: This suggestion was introduced in lines 26.
4.- Lines 45. Comparative adjectives such as “slower” cannot close a sentence. It needs an additional object to establish the comparison. For instance, “X is slower than Y.”
Answer: sentence was re-written in line 41:
“Interestingly, the conversion of EPA to produce DHA is slower than that of ALA into EPA [9].”
The
5.- Lines 53-54. “specific pro-resolving mediators” Why in italics?
Answer: The term was in italics because of its importance related to the text, however it does not have any other relevance, and now the term is not in italics (lines 49-50).
6.- Lines 55-57. “This review addresses only the resolvins, protectins and maresins, due to the solid information on these mediators available at the present time.” The abstract section states that DHA, resolvins, protectins and maresins are considered. However, according to the quoted statement DHA is not discussed. I suggest amending the abstract section accordingly. Also, I suggest rephrasing and improving the scientific style of the statement in question.
Answer: DHA was eliminated in line 14.
7.- Line 62. The authors state “cyclooxygenases”, however the role of these enzymes on the production of SPMs is not clear throughout the manuscript. In addition, the only instance where these enzymes are mentioned is to indicate just COX-2 (lines 94 and 216). It seems that other COXs have not involvement in SPMs production.
Answer: In agreement with this comment, the following paragraph was added in line 103:
“Moreover, the cyclooxygenases (COXs) are heme-containing enzymes that convert arachidonic acid to prostaglandin H (PGH), which are transformed in thromboxane A2 and prostacyclin. There are two major isoforms of COX, (i) COX-1 that is a constitutive enzyme widely distributed throughout the body and believed to play a maintenance or protective role; and (ii) COX-2 that is an inducible enzyme, whose levels and activity can increase rapidly and significantly in response to stimuli such as inflammatory mediators, thus being associated with inflammatory processes [35].”
New reference 35 was added to the list, and the rest of them were re-numbered in text and list.
Methods
8.- Line 72. “2010 onwards, including” instead of “2010 on including”
Answer: This sentence was corrected as indicated by the referee, in lines 73-74.
Synthesis and metabolism of DHA-derived lipid mediators
9.- Line 106. “The anti-inflammatory functions of SPMs favoring resolution are attained…” Stating anti-inflammatory function favouring resolution is redundant. I suggest “The anti-inflammatory functions are attained…”
Answer: The sentence outlined was re-written as suggested by the referee in line 115.
10.- Line 127. “In contrast, chronic inflammation…” instead of “Contrarily to acute inflammation, chronic inflammation…”
Answer: The sentence was re-written as indicated by the referee in line 136.
11.- Lines 145-148. “All of the above is of particular interest, considering that the traditional therapeutic approaches they have focused to suppress, block or inhibit the pro-inflammatory mediators, and, although several of them are effective, they can provoke immunologic suppression and infections [30].” What is “All of the above”? Who are “they” or them? The quoted statement is difficult to digest. Could you please rephrase it in a comprehensive, clear, grammatically well-constructed and more scientific style
Answer: According to the suggestion of the referee, the indicated paragraph was re-written in lines 153-156:
“Importantly, although the traditional therapeutic approaches have effectively focused to suppress, block or inhibit the pro-inflammatory mediators, these methodologies can provoke immunologic suppression and infections [33].”
New reference 33 was added to the list, and the rest of them were re-numbered in text and list.
11.- Line 171. “In agreement these findings…” grammatically incorrect.
Answer: This phrase was replaced in line 189 by:
“In agreement with these findings”
12.- Line 196. “tthrefore,…” delete extra letter
Answer: Done in line 211.
13.- Line 199. “…reservation of thedendritic tree” What does the quoted statement mean?
Answer: This was corrected in line 214:
“preservation of the dendritic tree”
Respiratory diseases
14.- It is incorrect to start this section by describing one of the several existing respiratory diseases. I suggest starting by explaining that SPMs have been used for the treatment of respiratory diseases and after that the authors can elaborate on the different respiratory diseases. Also, the authors can start by giving a more general statement on respiratory diseases as in section 7 for metabolic syndrome.
Answer:
Thanks for this comment. We improved the redaction in this section lines 222-226:
“SPMs specialized in airway inflammatory response have been used for the treatment of respiratory disease [88]. In this regard, respiratory diseases include a wide range of pathologies with different clinical manifestations, affecting the normal airways and lung function. An increase in the inflammatory response is considered a characteristic point of these diseases, being also a critical factor for their progression [88].”
New reference 88 was added to the reference list, and the rest were re-numbered in text and list.
15.- Lines 233-234. “It has been reported that inflammation and lung injury were persistent in pneumonia induced by Haemophilus influenzae in rats [98]” instead of “In pneumonia induced by Haemophilus influenzae in rats, Croasdell et al. found that inflammation and lung injury were persistent [98]”:
Answer: This change was carried out in lines 253-254 (ref. 98 changed to 104).
Metabolic syndrome
16.- Lines 253-253. “. In addition to the above data” What data are you referring to?
Answer: In order to clarify this point, “In addition to the above data” was changed to:
”In addition, treatment…” in line 271.
Cardiovascular diseases
17.- Line 281. According to the Oxford and Cambridge dictionaries, the expression “on the other hand” should be preceded by 'on the one hand'. Consequently, the authors should rephrase the whole paragraph.
Answer: The paragraph was rephrase to:
”Moreover, MaR1 decreased…” in line 300.
Liver diseases
18.- Lines 312-314. “… endoplasmic reticulum (ER) stress and lipogenic … decreases ER stress…” instead of “… endoplasmic reticulum ER stress and lipogenic … decreases ER stress…”
Answer: Done in line 331-332.
19.- Lines 327-328. “Research by our group evaluated the proposal that combined protocols involving n-3 PUFAs and other protecting agents may result in better responses than monotherapies…” This is an odd statement. I suggest rephrasing it. For example: “Our research group have suggested that combining n-3 PUFAs and other protecting agents may result in better responses than monotherapies…”
Answer: Done in lines 346-347.
20.- Lines 340-344. I suggest rephrasing as “Co-supplementation of the hormetic hormone L-3,3′ ,5-triiodothyronine (T3) with either (i) methylprednisolone, to preserve liver tissue regeneration post-hepatectomy [145]; or (ii) fish oil, to suppress ischemia-reperfusion inflammatory liver injury [146].” have been also suggested .” instead of “Other instances supporting the above proposition is the co-supplementation of the hormetic hormone L-3,3′ ,5-triiodothyronine (T3) with either (i) methylprednisolone, to preserve liver tissue regeneration post-hepatectomy [145]; or (ii) fish oil, to suppress ischemia-reperfusion inflammatory liver injury [146].”
Answer: Done in lines 359-362, and original refs. 145 and 146 were re-numbered into 151 and 152.
21.- Line 356. As mentioned above, the Oxford and Cambridge dictionaries state that the expression “on the other hand” should be preceded by 'on the one hand'. Consequently, the authors should rephrase the whole paragraph.
Answer: The indicated paragraph was rephrased into:
“A relationship has been also observed with some (i) dental pathologies,…” in line 375.
22.- References
Remove the double reference numbering. For example, 113, 114, 142-146
References 5, 38, 75, 87, 92, 133 should have space between year and volume. Check all the references.
Some references are given by capitalizing every first letter (e.g., 1, 3, 27, 29, etc) of the title, while other references only capitalized the first letter of the title (e.g. 2, 15, 16, etc). Try to be consistent.
Answer: All references were now written in the same style.
23.- Comments on the Quality of English Language
Very bad quality. Some sentences are too long and contain superfluous and repetitive information. The authors should improve the English level and use a more scientific
Answer: The level of English was improved in the revised manuscript.
Round 2
Reviewer 2 Report
The second revisions of this review (manuscript ID: nutrients-2491168) by Beyer, M.P.; Videla, L.A.; Farías, C.; and Valenzuela, R., address the comments brought up in the first round. However, the chemical structure of Maresin 2 in figure 2 is wrong. The authors should consult the literature or the Cayman Chemical website, one of the leading biotech companies selling this compound (https://www.caymanchem.com/search?q=mar2).
The authors should clarify in lines 95-99 that the referred to cysteinyl-containing SPMs are derived from DHA, not ARA or EPA; the sentence is confusing the authors are advised to break it down in shorter sentences--for clarity for the reader.
Reviewer 3 Report
The authors have addressed correctly, the issues mentioned in my first report